# Effects of Soil Arthropods on Non-Leaf Litter Decomposition: A Meta-Analysis

Wei Cheng [1], Liehua Tie [1,2,*], Shixing Zhou [2], Junxi Hu [2], Shengnan Ouyang [1,3] and Congde Huang [2]

1. Institute for Forest Resources and Environment of Guizhou, Key Laboratory of Forest Cultivation in Plateau Mountain of Guizhou Province, College of Forestry, Guizhou University, Guiyang 550025, China; chengwei20210917@163.com (W.C.); snouyang@gzu.edu.cn (S.O.)
2. National Forestry and Grassland Administration Key Laboratory of Forest Resources Conservation and Ecological Safety on the Upper Reaches of the Yangtze River, Sichuan Province Key Laboratory of Ecological Forestry Engineering on the Upper Reaches of the Yangtze River, College of Forestry, Sichuan Agricultural University, Chengdu 611130, China; szhou@sicau.edu.cn (S.Z.); junxihu@stu.sicau.edu.cn (J.H.); lyyxq100@aliyun.com (C.H.)
3. Swiss Federal Institute for Forest, Snow and Landscape Research, 8903 Birmensdorf, Switzerland
* Correspondence: lhtie@gzu.edu.cn; Tel.: +86-18-010-664-825

**Abstract:** According to the widely accepted triangle model, global litter decomposition is collectively controlled by climate, litter initial quality, and decomposers. However, the specific contribution of soil arthropods to litter, especially the non-leaf litter, the decomposition of terrestrial ecosystems and its drivers are still unclear. We conducted a global meta-analysis based on 268 pairs of data to determine the contribution and pattern of soil arthropods to branch, stem, and root litter decomposition in farmlands, forests, and grasslands and analyzed the relationship of soil arthropods' decomposition effect and potential drivers. Our results showed that: (1) soil arthropods increased global non-leaf litter mass loss by 32.3%; (2) the contribution varied with climate zone and ecosystem type, with a value of subtropical (53.3%) > temperate (18.7%) > tropical (14.7%) and of farmlands (40.6%) > grasslands (34.3%) > forests (0.6%), respectively; (3) the soil arthropods' decomposition effect gradually decreased with decomposition time, and it was higher in litterbags with a mesh size of 1–2 mm (65.4%) and >2 mm (49.8%) than that of 0.5–1 mm (13.6%); (4) the soil arthropods' decomposition effects were negatively correlated with the litter initial C/N ratio, mean annual precipitation (MAP; $p < 0.001$), and elevation and was positively correlated with litter weight. In conclusion, soil arthropod promoted global non-leaf litter decomposition, and the contribution varied with climate zone, ecosystem type, and decomposition time as well as litterbag mesh size. Overall, this study improves the understanding of soil arthropods driving global non-leaf litter decomposition.

**Keywords:** soil arthropod; litter decomposition; meta-analysis; biodegradation; mesh size





## 1. Introduction

More than 50% of plant productivity and nutrients in terrestrial ecosystems can be returned to soil via decomposing litter [1–3]. Therefore, litter decomposition plays a crucial role in terrestrial biogeochemical cycling [4]. The widely accepted triangle model claims that climate, litter initial quality, and decomposers are the drivers in controlling global litter decomposition [5]. It is estimated that climate and litter initial quality can explain approximately 60% to 70% of the total variation [6], while the contribution of soil biota is still uncertain. In fact, there are more than 25,000 species of soil arthropods with a biomass over 300 million tons in terrestrial ecosystems, and they are an important driver in global carbon (C) and nutrients cycling [7,8]. Although an increasing number of studies suggested that soil arthropods can regulate litter decomposition through a top–down effect [9], quantifying their specific contributions on a global scale is a great challenge and also a chance to improve the understand of soil biota function [10].

Prior studies quantified the contribution of soil faunas to global leaf litter decomposition through meta-analysis, while the results were inconsistent. For example, Xu et al. [11] reported that soil faunas increased the decomposition rate of leaf litter, especially in the litter with high initial cellulose content. Huang et al. [12] highlighted that the earthworms significantly promoted leaf litter decomposition from 80.9% to 107.7%. Peng et al. [13] reported that soil faunas increased the rate of leaf litter decomposition by 33.0%. However, Zan et al. [14] found that soil faunas decreased the leaf litter decomposition rate in tropical (200%), subtropical (47%), and temperate (28%) Chinese forests. Njoroge et al. [15] found that the presence of soil faunas enhanced leaf litter decomposition in the biomes with <2000 mm precipitation, while it has neutral effects in the biomes with >2000 mm precipitation. Moreover, a few studies have reported that the effects of soil faunas on the decomposition of dead wood. For instance, a field experiment across 55 study sites and six continents showed that soil faunas increased dead wood decomposition by 3.9% per year, but there were no significant effects in temperate and northern forests [16]. Nonetheless, the availability information of soil faunas' (especially soil arthropods) effects on global non-leaf litter (e.g., roots, branches, stem, etc.) decomposition is limited.

In addition, the results of field control experiments indicated that the effects of soil arthropods on the decomposition of non-leaf litter varied with ecosystem types. For example, Fujii et al. [17] found that soil arthropods decreased the litter mass loss in a forest at the Kyoto University Experimental Forestry Station. However, Justin et al. [18] found that soil arthropods promoted the decomposition of litter in a grassland ecosystem. Milcu et al. [19] also found that soil arthropods increased litter decomposition in grassland and farmland ecosystems. Schmidt et al. [20] reported that soil arthropods promoted straw litter decomposition in a tropical agroecosystem and the effect increased with increasing numbers of soil arthropods. Interestingly, Cassani et al. [21] found that the positive effect of soil arthropods on litter decomposition was stronger in grassland ecosystems (the increased decomposition rate was 0.0033 g/year) than that in agricultural ecosystems (the increased decomposition rate was 0.0024 g/year). Increasing studies also highlighted that the effect of soil arthropods on litter decomposition was different across climate zones. For instance, Fujii et al. [17] found that soil arthropods decreased litter decomposition in a temperate forest, while Liu et al. [22] found that soil faunas increased litter decomposition in a tropical forest. Moreover, Cassani et al. [23] reported that the soil arthropods increased the rate of litter decomposition during the first 6 months and did not affect the decomposition after 11 months of decomposition. Tian et al. [24] conducted an experiment using litterbags with 7 mm, 2 mm, and 0.5 mm mesh sizes, and they found that the decomposition rates of maize straw and rice straw litter were positively correlated with the mesh sizes. These results of the prior studies highlighted that the effect of soil arthropods on non-leaf litter decomposition most likely varied not only with climate zone and ecosystem type but also with decomposition time and litterbag mesh size.

Here, in order to determine the specific contribution of soil arthropods to non-leaf litter decomposition, we collected publications reporting soil arthropods' impact on root, branch, and stem litter decomposition using litterbags and then conducted a global meta-analysis. To determine the mechanism driving the effect of soil arthropods on global non-leaf litter decomposition, we also analyzed the correlations among the soil arthropods' decomposition effect and the potential predictor variables (i.e., litter initial C/N ratio, litter weight, mean annual precipitation (MAP), and elevation). We test the following hypotheses: (1) the presence of soil arthropods overall would promote the decomposition of non-leaf litter; (2) the soil arthropods' decomposition effect would decrease with increasing decomposition time, and the effect would vary with climate zone and ecosystem type; (3) the rate of non-leaf litter decomposition would be higher in larger mesh size litterbags than that in smaller ones, because a larger mesh size allows more soil arthropods to enter the litterbag.



## 2. Materials and Methods

### 2.1. Data Collection

The thesis, peer-reviewed articles, and published articles were searched on 1 March 2023 in Web of Science (https://www.webofscience.com) and China National Knowledge Infrastructure (https://www.cnki.net) using two search terms, i.e., soil fauna OR soil invertebrate OR arthropod OR collembola OR acarid OR nematode OR earthworm OR ant OR arachnid; litter decomposition OR litter mass loss OR litter remaining mass OR litter nutrient release; and their equivalents in Chinese. A total of 3249 studies were found with 2861 papers coming from Web of Science and 388 papers coming from China National Knowledge Infrastructure. These papers were from 1929 to 2023.

The literature screening was carried out based on the following criteria: (1) studies must focus on the impact of soil arthropods on litter decomposition and the study materials must include at least one of root, stem, and branch litter; (2) experiments must be using a litterbag and must inform the mesh size; (3) experiments must at least include the control group and treatment group; (4) studies must explicitly report the location and sampling time; and (5) for studies involving multiple sampling times, the values of variables at each sampling times must be provided. Based on the above conditions and after removing duplicates, we numbered each study uniformly in order to treat each study area as a separate study site. Then, the basic information of the literature was recorded, mainly including the author, country, journal, title, DOI, page range, latitude, and longitude. If there is no relevant information in the text, it is necessary to go to related literature to review and collect it as well as download the graphical information of each piece of literature to provide help for subsequent data extraction and analysis. Finally, a total of 268 pairs of data were extracted in our database. Among them, 47, 110, and 111 pairs of data were coming from tropical, subtropical, and temperate, respectively; 103, 91, and 62 pairs of data were coming from farmland, grassland, and forest ecosystems, respectively; 179, 82, and 7 pairs of data were during the decomposition time of 0–6, 6–12, and 12–24 months, respectively; and 123, 39, and 106 pairs of data were within litterbags of 0.5–1, 1–2, and >2 mm mesh size, respectively. The study sites in our database come from 7°30′ N to 51°25′ N and from 0°35′ W–123°4′ W to 4°50′ E–145°43′ E, mainly in Asia, Europe, and Africa. The elevations ranged from 50 to 2691 m and decomposition materials included roots, stems, and branches. More information in the screened literature can be found in Table S1, and the study sites are presented in Figure S1 (Supplementary Materials).

### 2.2. Data Extraction and Calculation

We focused on measuring litter mass loss as the key variable to determine the non-leaf litter decomposition responses to soil arthropods. We also collected information on several factors that potentially influence the soil arthropods' decomposition effects. These factors included the climate zone, the type of ecosystem, the time of decomposition experiment, the mesh size of the litterbag, the litter's initial C/N ratio, the MAP, the litter weight putting into the litterbags, and the elevation of each study site. By analyzing the soil arthropods' decomposition effects in different climate zones, ecosystem types, decomposition time, and mesh sizes, we hoped to determine the global pattern of the specific contribution of soil arthropods to non-leaf litter decomposition. By analyzing the correlation among soil arthropods' decomposition effects and the litter initial C/N ratio, litter weight, MAP, and elevation, we hoped to reveal the mechanism driving the effect of soil arthropods on global non-leaf litter decomposition.

The numerical data in the literature were recorded directly. The website data in the attachments were downloaded and then recorded. Data in figures were extracted using WebPlotDigitizer-4.2 on 10 March 2023 (https://automeris.io/WebPlotDigitizer). We quantify the contribution of soil arthropods to decomposition using litter mass loss (%) in different mesh sizes, which is converted from remaining mass (%) given in the literature, i.e., litter mass loss (%) = 100 − remaining mass (%). In addition, the climate zone, ecosystem type, litterbag or litterbox mesh size, experimental duration, elevation,

mean annual temperature (MAT), MAP, litter weight, and litter initial C/N (based on mass) were also extracted.

Standard deviation (SD) is used uniformly in this study. Therefore, the standard error (SE) was converted to SD using Equation (1):

$$SD = SE\sqrt{n} \tag{1}$$

where n is the number of repetitions.

To quantify the overall effect of soil arthropod on non-leaf litter decomposition by comparing litterbags or litterboxes including and excluding soil arthropods, we used the natural logarithm of the response ratio (ln(RR)) as the proxy of effect size [11,13]. The individual ln(RR) was calculated using Equation (2):

$$\text{In(RR)} = \text{In}\left(\frac{\bar{X_t}}{\bar{X_c}}\right) \tag{2}$$

where $(\bar{X_t})$ and $(\bar{X_c})$ were the mean values of litter mass loss in the treatment group and control group, respectively.

Then, the variance (V) associated with each ln(RR) was calculated using Equation (3):

$$V = \frac{S_t^2}{n_t \bar{x_t^2}} + \frac{S_c^2}{n_c \bar{x_c^2}} \tag{3}$$

where $S_t$ and $n_t$ were the SD and sample size of the treatment group, respectively; $S_c$ and $n_c$ were the standard deviation and sample size of the control group, respectively.

### 2.3. Statistical Analysis

In order to quantify the effect of soil arthropods on non-leaf litter decomposition, we used the "ram.mv" function in the "metafor" R package (R 4.2.3)to calculate the weighted mean effect sizes (lnRR$_{++}$) with 95% confidence intervals (CIs) [25]. If the lnRR$_{++}$ and 95% CIs do not overlap with the x-axis, it indicates a significant effect (i.e., increase or decrease) of soil arthropods on non-leaf litter decomposition [26]. To aid in interpreting, the lnRR$_{++}$ and 95% CIs were back-transformed using the equation ($e^{lnRR^{++}} - 1) \times 100$ [26]. In addition, the data were divided into subgroups, including (1) ecosystem type, including forest, farmland, and grassland ecosystems; (2) climate zone, including tropical, subtropical, and temperate zones; (3) decomposition time, including 0–6, 6–12, and 12–24 months; and (4) mesh size, including 0.5–1, 1–2, and >2 mm. The results of 95% CIs, d.f., and *Q*-value based on subcomponent grouping and test for heterogeneity are presented in Table S2 (Supplementary Materials).

To clarify the main variables driving the soil arthropods' decomposition effects, we applied the Akaike information criterion (AIC) model analysis in the "glmulti" R package to determine the relative importance of potential predictors (i.e., MAP, elevation, litter weight, and litter initial C/N ratio). The critical value of relative importance (sum of the Akaike weights) was set at 0.8 to distinguish necessary and unnecessary predictors, as reported by the previous study [27]. To determine the relationship between the soil arthropods' decomposition effect and its predictors, we furtherly performed linear regression analyses using the "geom_point" and "geom_smooth" functions in the "ggplot2" R package to determine the correlation of ln(RR) with ln(MAP), elevation, litter weight, and litter initial C/N ratio. All statistical analyses in this study were conducted using R software version 4.2.3 (R Core Team, Vienna, Austria, 2023).

## 3. Results

### 3.1. Effects of Soil Arthropods on Non-Leaf Litter Mass Loss

Overall, the presence of soil arthropods significantly increased non-leaf litter mass loss by 32.3% ($p < 0.001$; Figure 1). In addition, the results of subgroup analysis showed that climate zone, ecosystem type, decomposition time, and mesh size significantly affected the soil arthropods' decomposition effects ($p < 0.001$). Specifically, soil arthropods significantly increased the non-leaf litter mass loss in temperate, subtropical, and tropical biomes, with increases of 18.7% ($p < 0.001$), 53.3% ($p < 0.001$), and 14.7% ($p < 0.001$), respectively. The soil arthropods' decomposition effects in farmlands and grasslands were 40.6% ($p < 0.001$) and 34.3% ($p < 0.001$), respectively, and it was not significant in forests ($p > 0.05$). The soil arthropods' decomposition effects were 41.6% ($p < 0.001$), 18.6% ($p < 0.001$), and 10.9% ($p < 0.001$) during 0–6, 6–12, and 12–24 months of decomposition, respectively, suggesting the soil arthropods' decomposition effects decreased with increasing litter decomposition time. Moreover, the presence of soil arthropods significantly increased non-leaf litter mass loss within 0.5–1, 1–2, and >2 mm mesh size by 13.6% ($p < 0.001$), 65.4% ($p < 0.001$), and 49.8% ($p < 0.001$), respectively.

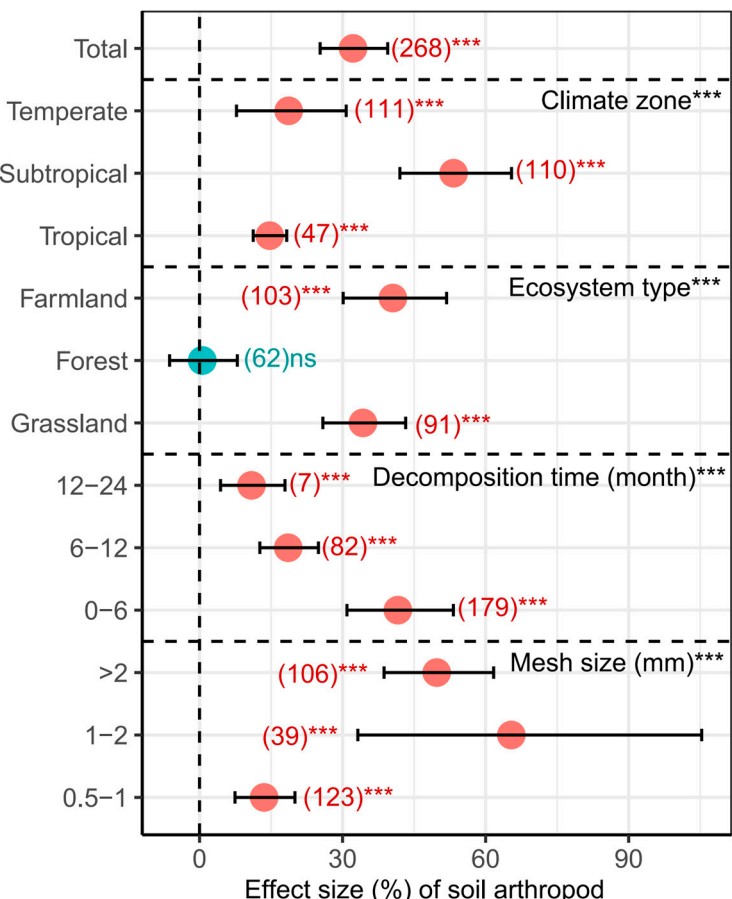

**Figure 1.** Effect size and 95% confidence intervals (CIs) of soil arthropods effects on non-leaf litter mass loss. The red font number in the parentheses indicates the number of data bars; ns indicates not significant; *** indicates $p < 0.001$.

### 3.2. Relationship between the Soil Arthropods' Decomposition Effect and Its Potential Drivers

The results of AIC showed that the litter initial C/N ratio and MAP were necessary predictors as their value of relative importance exceeded the set value (0.8) (Figure 2a), while elevation and litter weight were unnecessary predictors as their values of relative importance were lower than 0.8. In addition, the regression analysis results showed that the ln(RR) of mass loss was negatively line correlated with the litter initial C/N ratio,

MAP, and elevation ($p < 0.05$, Figure 2b–d), and it was positively line correlated with litter weight ($p < 0.001$; Figure 2e). Our results showed that the litter initial C/N ratio and MAP were important predictors driving the effect of soil arthropods on global non-leaf litter decomposition.

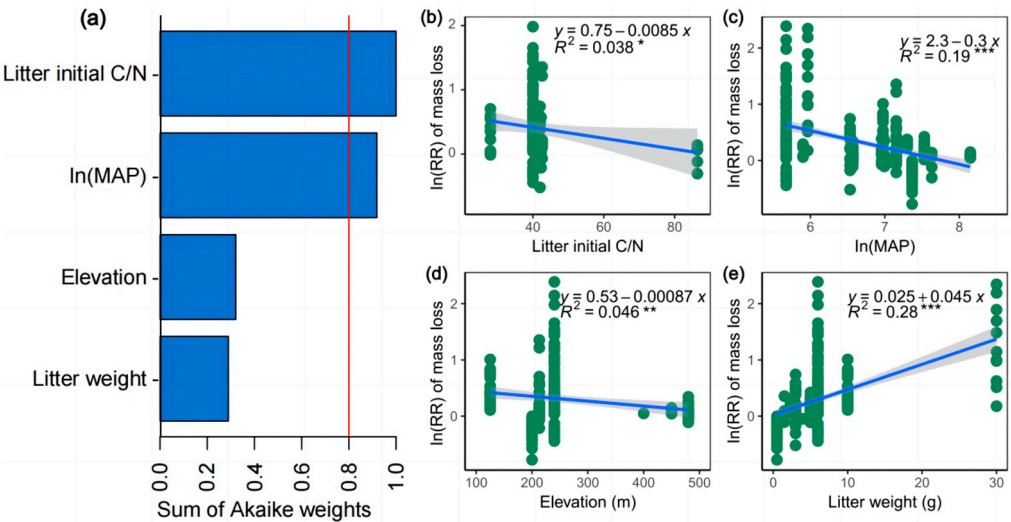

**Figure 2.** Potential predictors. (**a**) the relative importance of potential predictors based on the Akaike information criterion (AIC); the critical value is using 0.8 (red line) to distinguish the necessary and unnecessary predictors; (**b**–**e**) results of fitted linear regression analysis lines and 95% confidence intervals; *, **, and *** indicate $p < 0.05$, 0.01, and 0.001, respectively.

## 4. Discussion

Our results showed that the presence of soil arthropods stimulated the decomposition of global non-leaf litter by 32.3% (Figure 1), supporting our fist hypothesis, which was also consistent with prior meta-analysis studies reporting soil faunas increased global leaf litter decomposition rate by 30.9%~37% [11,28]. This highlighted that soil arthropods were one of the crucial drivers not only for global leaf litter decomposition but also for global non-leaf litter decomposition. We, therefore, strongly suggested taking soil arthropods' decomposition effects into account in predicting global litter decomposition to improve the understanding of C and nutrient cycling of terrestrial ecosystems.

The triangular model claims that climate (e.g., MAP) is a key controlling global leaf litter decomposition [4]. Our study found that climate zone influenced soil arthropods' decomposition effects, which is consistent with our second hypothesis. Interestingly, we found lower soil arthropods' decomposition effects in the tropics than in the subtropics, which is inconsistent with Wall et al. [29], who reported that soil faunal effects are higher in warmer locations. There are two possible reasons. First, in our study, the soil arthropods' decomposition effects were lower in forests than that in grasslands and farmlands, which will be explained in the next paragraph. In our database, the percentage of forests in the tropical regions was 12.8% and this value was 5.2% in the subtropical regions, resulting in a lower soil arthropods' decomposition effect in tropical regions than that in subtropical regions. Second, the results of our regression analysis showed that the soil arthropods' decomposition effect was negatively correlated with MAP, which can explain the lower soil arthropods' decomposition effect in tropical ecosystems because tropical regions often have higher MAP than that in subtropical regions. Therefore, it is necessary to increase the experimental data of soil arthropods on non-leaf litter decomposition in the future to make up for the deficiency of this study.

Our findings show that soil arthropods have variable effects on non-leaf litter decomposition across biomes, supporting our second hypothesis that their decomposition effect varies among habitats. Specifically, soil arthropods promoted non-leaf litter decomposition

in both farmlands and grasslands, which was consistent with prior studies [21,30]. For example, Castro-Huerta et al. [31] reported that soil faunas increased the decomposition of grass root litter by 14.4%. Schmidt et al. [32] also found that the presence of soil arthropods increased the decomposition rate of straw litter by 45% in agricultural ecosystems. Interestingly, the soil arthropods' decomposition effect in forests was not significant in our study, which was inconsistent with the results of prior study [13]. This phenomenon may be related to the litter initial C/N ratio, since our results of regression analysis and AIC showed that the litter initial C/N ratio was negatively correlated with the soil arthropods' decomposition effect (Figure 2a,b). As a key driver controlling litter decomposition, a higher litter initial C/N ratio indicates a lower quality and palatability of litter [22,33]. The feeding activity and decomposition effect of soil arthropods, therefore, generally decreases with increasing litter initial C/N ratio and vice versa [34–38]. The litter initial C/N ratio in our database was 38.9 ± 3.36, 40.4 ± 0.71, and 64.2 ± 13.2 in grasslands, farmlands, and forests, respectively. This can explain why the soil arthropods' decomposition effect was positive in farmlands and grasslands, while it was not significant in forests in our study. Frouz et al. [39] also reported that soil arthropods increased the decomposition rate of litter with an initial C/N ratio < 50; otherwise (>50), there was a neutral or even negative effect, which supported our results. Taken together, we suggest that the litter initial C/N ratio as well as litter resources palatability should be considered in the future to improve the accuracy of assessing the effect of soil arthropods on global non-leaf litter decomposition.

Our study results showed that soil arthropods had a stronger effect at the early stage than that at the later stage, supporting the second hypothesis pointing out that the soil arthropods' decomposition effect would decrease with increasing decomposition time, which was consistent with prior studies [40]. Tie et al. [41] showed that the decomposition rate of litter was faster at the early stages of decomposition (the first 6 months) than at the middle and late stages of decomposition (6–12 months). The decreased soil arthropods' decomposition effect can be explained by two potential mechanisms. First, the easily degradable organic matter and nutrients (e.g., K and Na) can release from litter quickly, which may result in an increase in the relative content of recalcitrant components (e.g., lignin) and C [42]. Soil arthropods prefer to eat the litter resources with higher nutrients (e.g., N) rather than higher C; thus, their effect may be higher at the early stage than that at the later stage [43,44]. Second, litter decomposition mediated by soil arthropods usually leads to a short-term increase in microbial activity in feces, while this activity decreases in such feces at the long term [39], resulting in a decrease biodegradation with increasing decomposition time. All in all, we recommend introducing soil arthropods into grasslands and farmlands to decompose fresh fallen root, branch, and stem and thus to accelerate the transfer of C and nutrients (e.g., N, K, and Na) from litter to soil, but additional management measures may have to be implemented in order to continuously improve decomposed litter degradation at the later stage.

In our study, the soil arthropods' decomposition effect was higher in 1–2 mm and >2 mm litterbags than that in 0.5–1 mm, which was similar with the results of prior studies reporting that the decomposition rate within a 2 mm mesh size litterbag was higher than that within 0.01 and 0.08 mm [45,46]. However, in this study, there is no difference between the effects of the 1–2 mm and >2 mm sizes. This result may be related to the trophic structure of soil arthropods within litterbags. Soil arthropods can regulate the decomposition of litter via a top–down effect, including the following pathways [9,40]. First, herbivorous soil arthropods (i.e., primary decomposers) directly affect litter decomposition through feeding and fragmentation [39]. Second, bacteriovorous and fungivorous soil arthropods (i.e., secondary decomposers) indirectly affect litter decomposition by selectively feeding on bacteria and fungi, respectively [9,47]. Third, predatory soil arthropods (i.e., predators) indirectly affect litter decomposition by preying on the primary and/or secondary decomposers [9,39]. The influence of soil arthropods on litter decomposition, therefore, is not only affected by their quantity and diversity but also by their trophic structure [48]. Although a >2 mm mesh size allows soil macro-fauna (e.g., spider, the predator) to enter litterbags,

they can also affect meso- and micro-fauna (e.g., collembola, the decomposer) diversity and structure [47]. This can explain why the soil arthropods' decomposition effect in our study was not significant between the 1–2 mm and >2 mm litterbags. In addition, part of the litter will transform into feces after passing through the intestines of soil arthropods [48]. The contents of nutrients (e.g., N and P) and C in feces are usually far from that in plant litter because the ability of soil arthropods absorbing nutrients (e.g., N and P) and C is different [49], which increased the uncertainty of the soil arthropod decomposition effect in litterbags [50]. Overall, our third hypothesis was partially supported. Determining the effect of the trophic structure and feces of soil arthropods on litter decomposition provides a big chance to furtherly reveal the mechanism of global litter decomposition responses to soil arthropods. We, thus, predict this is most likely to be one of the hot topics of global litter biodegradation in the future.

This study quantified the specific contribution of soil arthropods to the non-leaf litter decomposition of terrestrial ecosystems and clarified the differences of soil arthropods' decomposition effects across climate zones, ecosystem types, decomposition time, and mesh sizes. In addition, we determined that the litter initial C/N ratio and MAP were the necessary predictors controlling the effects of soil arthropods on global non-leaf litter decomposition. In the future, we can consider the following scientific issues to furtherly improve the understanding of the mechanism of soil arthropods driving global litter decomposition. First, the soil arthropods' decomposition effects were negatively correlated with non-leaf litter initial C/N ratio, indicating that nutrients' availability (e.g., N) most likely regulated the response of global litter C decomposition to soil arthropods. Therefore, in the future, we can try to study the global litter biodegradation pattern from the perspective of soil arthropods-mediated stoichiometric decomposition. Second, our results highlighted that the soil arthropods' decomposition effects were also negatively correlated with the MAP, which was similar with the study by Njoroge et al. [15] who reported that the soil faunal decomposition effect was higher in dry environments. We speculate that this may be due to the differences of soil faunal diversity under different precipitation. However, the available information is still limited. Thus, more studies should be conducted to clarify the soil arthropod diversity under specific precipitation and determine its impact on global litter decomposition. Finally, our database included the studies in Asia, Europe, and Africa, while the studies in North America, South America, Oceania, and Antarctica are still lacking. In the future, we suggest that more researchers consider the non-leaf litter decomposition responses to soil arthropods, especially in those regions now lacking studies, in order to describe the worldwide pattern of soil arthropods' decomposition effects. All in all, our results indicated that soil arthropods overall increased non-leaf litter mass loss, while the increases varied with ecosystem type, climatic zone, decomposition time, and mesh size.

## 5. Conclusions

Soil arthropods increased the global non-leaf litter mass loss by 32.3%, but this effect varied with climatic zone, ecosystem type, decomposition time, and litterbag mesh size. The results highlighted that soil arthropods were one of crucial drivers for global non-leaf litter decomposition. Therefore, we suggest that soil arthropods should be included in global non-leaf litter decomposition models to improve the accuracy of C and nutrients' cycle prediction. Additionally, our result showed that the soil arthropods' decomposition effects were negatively correlated with litter initial C/N ratio and MAP, which was expected within the framework of the triangle model. Overall, this study quantified the contribution of soil arthropods to global non-leaf litter decomposition and improved our understanding of soil arthropods' decomposition effect.

**Supplementary Materials:** The following supporting information can be downloaded at: http://www.mdpi.com/xxx/s1, Table S1: Geographical distribution of the experimental sites in this study; Table S2: The 95% confidence interval (CI), d.f., and Q-value based on subcomponent grouping and test for heterogeneity; Figure S1: Information list of the literatures selected in our database.

**Author Contributions:** Conceptualization, L.T. and W.C.; software, W.C., L.T. and J.H.; data curation, W.C. and L.T.; writing—original draft preparation, W.C. and L.T.; writing—review and editing, L.T., S.Z., J.H., S.O. and C.H.; visualization, W.C.; supervision, L.T. and S.O.; project administration, L.T. and S.O.; funding acquisition, L.T. and S.O. All authors have read and agreed to the published version of the manuscript.

**Funding:** This research was funded by the Guizhou Provincial Science and Technology Projects (ZK[2022]YIBAN101; ZK[2023]YIBAN110) and the Natural Science Project of Guizhou University (2021-31 and 2022-27).

**Data Availability Statement:** The data in this study are available from the authors upon request.

**Acknowledgments:** We gratefully thank Chengming You and Jie Wang for their valuable suggestions for improving the manuscript.

**Conflicts of Interest:** The authors declare no conflict of interest.

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
