# Peer review of "Effects of Soil Arthropods on Non-Leaf Litter Decomposition: A Meta-Analysis"

_forests, doi:10.3390/f14081557_

Round 1

Reviewer 1 Report

The paper show a interesting meta-anlysis of the role of arthopods effect on non litter decomposition process. This is an interesting subject and auhtor show some data about that, nevertheless can be add some papers to improve the introduction. 

Authors may be explain the selection the time period selected to the review. 

Data analysis are property shown and results are interesting, but can considered some aspects as the effect of microenvironmente in mesh size in the litter bags studies and also the effecto of the microbiota of the arthropods feces in the own decomposition process.

Reviewer 2 Report

These are my main comments on the manuscript (forests-2507020) entitled “Effects of soil arthropods on non-leaf litter decomposition: a meta-analysis”. This work investigates global meta-analysis to determine the contribution and pattern of soil arthropods to branch, stem, and root litter decomposition in farmlands, forests, and grasslands and analyzed the relationship of soil arthropods decomposition effect and potential drivers. Following substantial revisions should be incorporated in the manuscript.

1. I have concerns about the manuscript sections that I believe need to be addressed in order to improve its clarity.

2. In results section, statistical data are needed. Please, provide the X2 value and IC values, degree freedom, and P-value obtained from statistical analysis.

3. Other revisions could be checked in PDF attached.

Discussion section should be summarized in all lines
